# Prediction of Concrete Strength with P-, S-, R-Wave Velocities by Support Vector Machine (SVM) and Artificial Neural Network (ANN)

**Jong Yil Park [1], Young Geun Yoon [2],\* and Tae Keun Oh [2,3],\***

1    Department of Safety Engineering, Seoul National University of Science and Technology, Seoul 01811, Korea; jip111@seoultech.ac.kr
2    Department of Safety Engineering, Incheon National University, Incheon 22012, Korea
3    Research Institute for Engineering and Technology, Incheon National University, Incheon 22012, Korea
\*    Correspondence: y_young_geun@naver.com (Y.G.Y.); tkoh@inu.ac.kr (T.K.O.); Tel.: +82-032-835-8294

**Abstract:** Mechanical waves, such as ultrasonic waves, have shown promise for use in non-destructive methods used in the evaluation of concrete properties, such as strength and elasticity. However, accurate estimation of the concrete compressive strength is difficult if only the pressure waves (P-waves) are considered, which is common in non-destructive methods. P-waves cannot reflect various factors such as the types of aggregates and cement, the fine aggregate modulus, and the interfacial transition zone, influencing the concrete strength. In this study, shear waves (S-waves) and Rayleigh waves (R-waves) were additionally used to obtain a more accurate prediction of the concrete strength. The velocities of three types of mechanical waves were measured by recent ultrasonic testing methods. Two machine learning models—a support vector machine (SVM) and an artificial neural network (ANN)—were developed within the MATLAB programming environment. Both models were successfully used to model the relationship between the mechanical wave velocities and the concrete compressive strength. The machine learning model that included the P-, S-, and R-wave velocities was more accurate than the model that included only the P-wave velocity.

**Keywords:** ultrasound; P-wave; S-wave; R-wave; non-destructive method; concrete strength; support vector machine; artificial neural network

## 1. Introduction

To evaluate the compressive strength of concrete in cast-in-place or existing structures, the standard destructive test is the most reliable method. Typically, standard cylindrical samples are made on site and sent to the laboratory for compressive strength testing. However, the test results may not be representative, because many factors such as the type and size of the aggregate and cement, the fine aggregate modulus, the water to cement ratio, and the interfacial transition zone, etc., are not considered [1]. Furthermore, the drilling of core samples from existing concrete members is not always possible on site and the concrete member might be damaged by the process. In this respect, one suitable alternative is to apply non-destructive evaluation (NDE) methods to estimate the concrete strength when core sampling is not preferable. The most widely used NDE method for the strength evaluation of concrete is the ultrasonic pulse velocity test, which measures the velocity between two transducers on both sides of a specimen. The velocity can be calculated based on the assumption that the wave path is known in advance. Many attempts to apply the pressure wave velocity Vp (km/s) as a measure of the concrete compressive strength have been made, using field convenience and simple equipment.

The estimation of concrete strength is typically based on various empirical relations between the concrete strength and nondestructive variables, such as the ultrasonic pulse velocity and rebound

number. NDE tests typically have their own empirical relations to connect the test parameter and concrete strength, but these depend strongly on factors such as the water to cement ratio, type and size of the aggregate and cement, and the amount of aggregate. Thus, they cannot be applied universally and require supplemental methods. Furthermore, the typical approach using regression analysis, such as with an exponential function, has not been successful [1].

The proposed equations have not been accurate for strength estimations because the strength may depend on factors other than the P-wave velocity (Vp). There are many factors that affect concrete strength, in addition to the ultrasonic pulse velocity. For example, the type and size of the aggregate affect the relationship between Vp and the concrete strength. Concrete with the largest aggregate content tends to have the highest pulse velocity [2]. Furthermore, a higher water content results in a higher ultrasonic velocity of the concrete [3,4]. However, a higher water content typically corresponds to a lower strength of the concrete [5]. This inconsistency can make the interpretation of the ultrasound results difficult. Trtnik et al. [6] investigated the effects of many factors, including the type, size, and shape of the aggregate, the concrete cast temperature, and the water to cement ratio, on the ultrasonic velocity. They concluded that the aggregate properties were the most effective factors.

For normal-strength concrete, many regression methods, such as linear or nonlinear regressions, have been applied to estimate the concrete strength. However, such regression methods cannot directly predict the concrete strength. Recently, various studies have been performed to obtain more accurate predictions of the concrete strength using machine learning algorithms, such as support vector machine (SVM) and artificial neural network (ANN) methods [7–10]. Prasad et al. analyzed the performance of an ANN to estimate the 28-day concrete strength of normal and high-strength concretes [9]. In their work, factors such as the water to cement ratio, aggregate to cement ratio, and the amount of cement were used as input parameters, and the concrete strength was the output of the network.

As an alternative to ANN, an SVM was developed that could effectively classify data and minimize the risk [11]. Unlike ANN, which operates based on the minimization of the training error, the SVM was created based on the minimization of the upper bound of the generalization error, which summarized the training error and confidential term [12]. The aim of the SVM method is to find the global optimum rather than local optima by solving the nonlinear problem in a high-dimensional region. Associated with an insensitive loss function, the SVM method was designed to process nonlinear regression problems, such as wind velocity estimation [13], traffic flow prediction [14], financial time-series prediction [15], and electricity load prediction [16]. In particular, the SVM has proven to be successful in the prediction of concrete strength. Yan and Shi [17] used an SVM model to accurately predict the strength and elasticity modulus of concrete. Ahmadi-Nedushan [18] also predicted these properties with the conventional method and a high-performance model using an adjustable fuzzy neural network, proving that this method is highly reliable. Yuvaraj et al. [19] predicted the fracture properties of concrete beams using an SVM model and compared them against experimental results. Gencel et al. [20] adopted an ANN and linear regression algorithm to study the abrasion resistance of concrete with different constituents. The results demonstrated that the ANN was more reliable than the conventional linear regression algorithm.

The capacity of the SVM depends mainly on a penalty factor, a kernel function parameter, and the width of an insensitive loss function. In this sense, the key to the application of the SVM to the prediction of concrete strength is to set the most appropriate parameters. Although previous researchers have tried to optimize the parameters, no approach has been able to provide optimal setting criteria.

This study began with an effort to determine major factors other than the P-wave velocity that are related to concrete strength. It is difficult to estimate the concrete strength indirectly from the mix design or existing buildings, and the predicted strength may not be accurate for newly constructed buildings. Typically, it is most reasonable to measure the current strength based on the physical information identified by reliable methods, such as the rebound hardness, the ultrasonic velocity, and electromagnetic waves. In this regard, the aim of this study was to increase the accuracy of strength

evaluations by incorporating the shear wave (S-wave) and Rayleigh wave (R-wave) velocities, because P-waves with small energies and waveforms may not yield accurate predictions of the concrete strength. From the measurements of S- and R-waves, it is possible to predict other properties, such as the shear modulus and Poisson's ratio, and to estimate the P-wave itself. Also, the S-wave is not affected by the water in concrete and R-wave has the advantage of being sensitive to concrete surfaces [2,3].

In this research, an artificial-intelligence-based approach is proposed that incorporates SVM and ANN models as alternatives to regression approaches [21–23]. In addition, more core samples were used to enhance the reliability of the analysis. Although fewer than 20 samples were used in most of the previous studies, 72 core samples were used in this study. The compressive strength tests were performed on 72 core samples, and three types of ultrasonic tests were performed before the core samples were produced. Based on the data, ANN and SVM models were developed using the MATLAB software, and concrete strengths were predicted more accurately than those predicted using the regression approach. Thus, the P-, S-, and R-wave ultrasonic pulse velocities were used as the SVM and ANN model inputs, and the actual compressive strengths from the destructive tests were used as the model output.

## 2. Ultrasonic Pulse Velocity Test

### 2.1. P-Wave Velocity (Vp) Measurement

The pulse velocities of pressure waves have been applied to measure the elastic modulus, strength and defects of concrete, by measuring a reflection wave or the velocity of an elastic wave in concrete [1]. The ultrasonic pulse velocity measurement can be used to assess the concrete quality characteristics, such as the density and uniformity, and to identify the presence of voids and cracks [24]. As the ultrasonic pulse velocity increases, the likelihood for the concrete to possess high quality (e.g., possessing no cracks), uniformity, and the designed density, is higher. Typically, this test uses an ultrasonic pulse with a 50-kHz center frequency transmitted from an electro-acoustic transducer that is placed in contact with one surface of the concrete. After passing the ultrasonic pulse, the other transducer placed on the opposite side detects the pulse. The elapsed time between both transducers is measured in milliseconds. Using the known traveling distance, D, and the measured travel time, T, the pulse velocity (V = D / T) can be calculated [25,26]. The ultrasonic tests in this study were based on the indirect method, as shown in Figure 1.

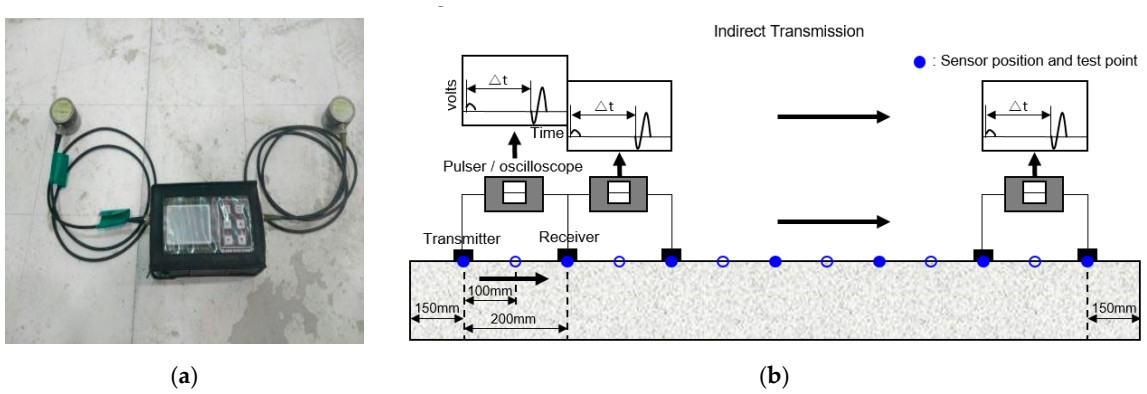

(**a**)　　　　　　　　　　　　　　　　(**b**)

**Figure 1.** Ultrasonic pulse velocity test method. (**a**) Ultrasonic equipment; (**b**) Procedure of P-wave measurement.

Using the test grid shown in Figure 2, the average velocity between two points in rows A–F was measured for each column (1–9). The "o" symbols denote the positions of the transducers, and the "x" symbols denote the average positions between adjacent points. In this test, access to the bottom surface was difficult, and indirect measurements were used on the surface. Six measurements were made for each column, and the average value was assumed to be the average P-wave velocity for this column.

For verification, a cylinder core (10 cm × 20 cm) in each column was extracted, and the compressive strength was measured according to the ASTM C 39 standard and used for comparative analysis. A total of 54 P-wave measurements were made, and nine specimens were taken for the compressive strength evaluation on each slab.

Because the surface of the concrete was not smooth, couplants, such as Vaseline, were used to facilitate the transmission of energy between the transducer and the surface. The experiments were carried out by two people. One person placed an ultrasonic sensor at each point, started it, and read the result. The other person recorded the results on a printed grid. A total of six slabs were tested in the same manner.

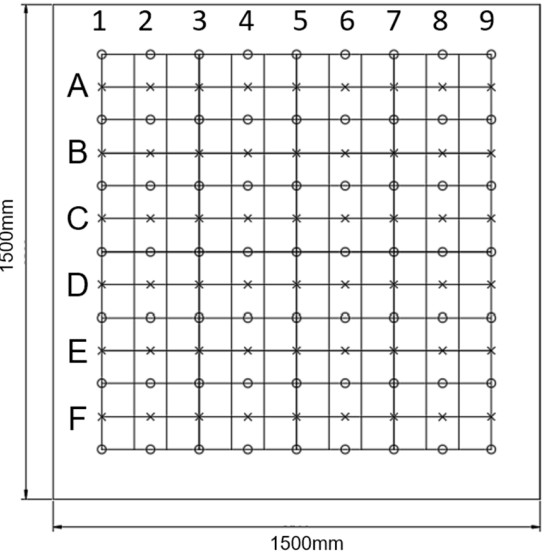

**Figure 2.** Test grid points on the slab.

## 2.2. S-Wave Velocity (Vs) Measurement

S-wave measurements were conducted using a Pundit 250 array with 36 sensors, as shown in Figure 3. The array used DPC (dry point contact) technology without a couplant, and 56 A scans were obtained with ultrasonic multi-channel pulse echo technology using shear waves, after which the real-time B scans were obtained. Shear waves cannot pass through liquids or gases, so they were not affected by the water in concrete. The same number of S-wave velocities were measured at the same positions (indicated by the "x" symbols in Figure 2) as the P-wave measurements. The measurements were performed five times at each point, and the average value was calculated. The experiment was carried out by two people. One person placed the equipment at each point and measured and read the results. The other person recorded the result on a printed grid. A total of six slabs were tested in the same manner.

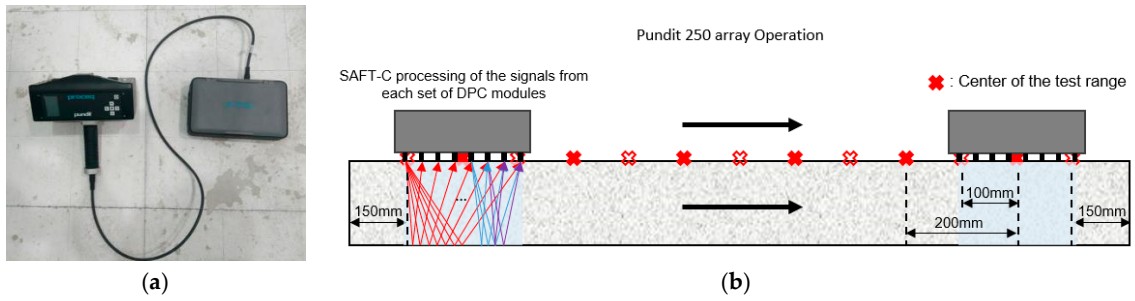

**Figure 3.** Shear wave tomography test method. (**a**) Tomography equipment; (**b**) Procedure of S-wave measurement.

### 2.3. R-Wave Velocity (Vr) Measurement

For the R-wave velocity measurements, the MASW (multi-channel analysis surface wave) associated with the induced mechanical waves passing through the concrete medium was measured by attaching a multi-sensor to the concrete surface, and generating and measuring the surface waves. The R-wave velocity was obtained by the phase velocity dispersion curve with high accuracy. A high signal to noise ratio (SNR) and good resolution were obtained by minimizing the noise using multiple sensors.

In this study, a total of eight sensors were used, and the distance between each sensor was 10 cm, as shown in Figure 4. Experiments were conducted by two people. One person placed and initiated the MASW equipment on each line in Figure 2, and the other hit the concrete surface using an impact hammer and saved the collected data in a Microsoft Excel file.

To obtain the surface wave velocity, the relationship between the velocity and frequency of the surface wave propagating in the target structure must first be clarified. In general, the relationship between the frequency and velocity of a wave is called a dispersion curve. To obtain the dispersion curve from the MASW test, data from multiple sensors is necessary, as shown in Figure 5. The surface wave velocity can be obtained by measuring the phase difference between each sensor. The obtained time-domain signal is represented as a phase velocity for various modes in the frequency domain, among which the A0 and S0 modes converged to the Rayleigh wave velocity.

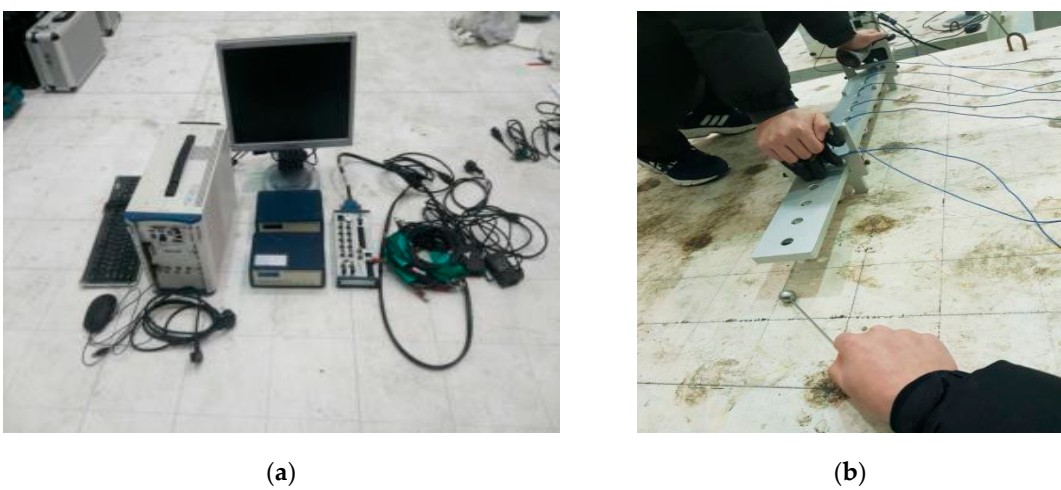

(**a**)       (**b**)

**Figure 4.** Test method for the multi-channel analysis surface wave (MASW). (**a**) Composition of test equipment; (**b**) Sensor placement for MASW test.

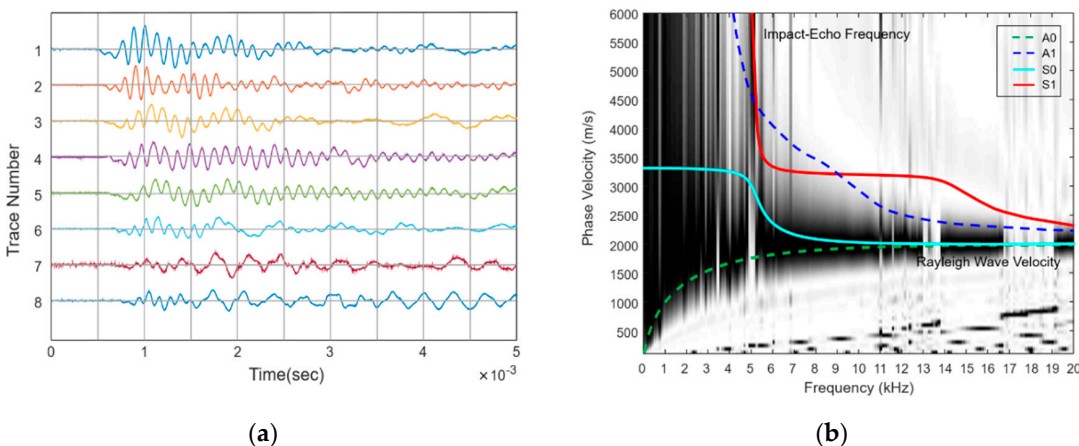

(**a**)       (**b**)

**Figure 5.** Use of the dispersion curve to determine the R-wave velocity. (**a**) Time domain; (**b**) Dispersion curve.

The relationships between the P-, S-, and R-wave velocities and the material properties are shown in Equations (1)–(3), respectively, assuming that the concrete was an elastic body. These equations can be used to estimate the elastic modulus, Poisson ratio, and density through the P-wave, S-wave, and R-wave velocities measured, using the methods described above. It was assumed that the density of concrete was 2400 kg/m$^3$ and the Poisson ratio was in the range of 0.2–0.3. The predicted elastic modulus and strength of the concrete were also similar to those measured by the ASTM C39 standard.

$$V_P = \sqrt{\frac{E(1-v)}{(1+v)(1-2v)\rho}} \tag{1}$$

$$V_S = \sqrt{\frac{E}{2(1+v)\rho}} \tag{2}$$

$$V_R = \frac{0.87 + 1.12v}{1+v} V_S \tag{3}$$

where, $E$, $v$, and $\rho$ are the elastic modulus, Poisson ratio, and density, respectively.

The goal of this study was to develop a machine learning model that can predict the concrete strength more accurately using P-, S-, and R-wave velocities within reasonable ranges.

## 3. Machine Learning Technique

### 3.1. Support Vector Machines and Model Development

SVM is a supervised learning method for regression and classification analysis based on risk minimization, first proposed by Vapnik [27]. The SVM method can process the classification of data using a kernel trick, implicitly mapping inputs into high-dimensional feature spaces. The SVM method uses an effective separation by a hyperplane, with the largest distance to the nearest training-data point and the lowest generalization error, allowing the SVM to obtain a better generalization. In comparison with other machine learning algorithms, the SVM method has several advantages, such as a unique optimization approach and the effective utilization of high-dimensional feature spaces and computational learning theory [28,29]. The mathematical algorithm is summarized below [30].

For a training dataset of n points of the form (*xi*, *di*), *xi* is the input vector, di is the target value, and n is the size of the dataset obtained by the mapping (φ) of *xi* into a high-dimensional feature space. This can be expressed as follows:

$$\text{f}(x) = \omega \cdot \phi(x) + \text{b} \tag{4}$$

where $\omega$ is the weight vector, $\phi$ is the high-dimensional feature space, and b is the bias of the hyperplane.

The basic concept of the SVM is to minimize the structural risks, and $\omega$ and b can be obtained by minimizing the following risk penalty function:

$$\text{Minimize } \frac{1}{2}\|\omega\|^2 \tag{5}$$

subject to

$$\begin{cases} d_i - \omega \cdot \phi(x) - \text{b} \leq \delta \\ \omega \cdot \phi(x) + \text{b} - d_i \leq \delta \end{cases} \tag{6}$$

where $\delta$ is the maximum deviation.

To determine the optimal values of $\omega$ and b, slack variables $\xi$ and $\xi^*$ are adopted, and the following new equivalent function is introduced:

$$\text{C}\frac{1}{n}\sum_{i=1}^{n}(\xi_i + \xi_i^*) + \frac{1}{2}\|\omega\|^2 \tag{7}$$

subject to:

$$\begin{cases} d_i - \omega\phi(x_i) - b_i \leq \varepsilon + \xi_i \\ \omega\phi(x_i) + b_i - d_i \leq \varepsilon + \xi_i^* \\ \xi_i \xi_i^* \geq 0 \end{cases} \tag{8}$$

where $C\frac{1}{n}\sum_{i=1}^{n}(\xi_i + \xi_i^*)$ is the estimated risk, $\frac{1}{2}\|\omega\|^2$ is the Euclidean norm as a penalty item, and C is the penalty constant used to control the penalty error.

Using the Lagrange multipliers, $a_i$ and $a_i^*$, the SVM function becomes:

$$f\left(x, a_i, \ a_i^*\right) = \sum_{i=1}^{n}\left(a_i - a_i^*\right)K(x, x_i) + b \tag{9}$$

Next, the Lagrange multipliers are included in the penalty objective function and the dual function can be obtained as follows:

Maximize

$$R\left(a_i, \ a_i^*\right) = \sum_{i=1}^{n} d_i\left(a_i - a_i^*\right) - \varepsilon(a_i - a_i^*\ ) - \frac{1}{2}\sum_{i=1}^{n}\sum_{j=1}^{n}\left(a_i - a_i^*\right)\left(a_j - a_j^*\right)K(x, x_i) \tag{10}$$

subject to:

$$\sum_{i=1}^{n}\left(a_i - a_i^*\right) = 0 \qquad \begin{array}{l} 0 \leq a_i \leq C \ \ i = 1, 2, \ldots n \\ 0 \leq a_i^* \leq C \ \ i = 1, 2, \ldots n \end{array} \tag{11}$$

$K(x_i, x_i)$ is the kernel function containing the inner product of $x_i$ and $x_j$, with corresponding feature spaces $\psi(x_i)$ and $\psi(x_j)$, respectively. Linear, polynomial, sigmoid, and Gaussian kernels are the most commonly used kernel functions. The Gaussian kernel is the most popular because it can reduce the complexity of the inputs effectively through $C$ and $\gamma$ [29]. In this study, the Gaussian kernel was applied for building the SVM classifiers. The Gaussian kernel was as follows:

$$K\left(x_i, x_j\right) = exp\left(-\gamma\left(x_i, x_j\right)^2\right) \tag{12}$$

where $\gamma$ is a constant.

The SVM model was implemented in the MATLAB environment using 'fitrsvm,' which trains an SVM regression model as a low-through moderate-dimensional predictor [30]. The SVM model supports the mapping of the predictor data using kernel functions [31]. It uses an adaptive genetic algorithm to optimize the $C$ and $\gamma$ values for the Gaussian kernel. Furthermore, it utilizes a least squares loss function, and thus, the problem is more efficiently managed by the Karush–Kuhn–Tucker (KKT) conditions [32,33].

The P-, S-, and R-wave velocities (Vp, Vs, and Vr) measured from the test samples were used as the model inputs, and the compressive strengths of the core samples were used as the model outputs. A total of 72 core cylinder samples were used for the SVM model. The datasets were first loaded into the input and output data, after which they were used for the model development and validation. The trained SVM regression model was subsequently used to generate the predictor values using the input data from the testing dataset. The predicted outputs were subsequently compared with the objective values, which were the compressive strengths for the validation.

### 3.2. Artificial Neural Network (ANN) and Model Development

An ANN is an information processing system modeled as a biological neural network, such as the human brain. An ANN utilizes connected artificial neurons, and its inherent behaviors can be explained by training the input parameters using the neurons, which results in a nonlinear mapping [34].

Rumerlhar et al. [35] first proposed the multilayer feed-forward back propagation network (MFBPN), one of the most popular ANN methods, owing to its simplicity and plasticity. The MFBPN consists of input, hidden, and output layers. In the ANN implementation, connections of all the artificial neurons in each layer consist of weights and biases. The neuron values from the previous layer are adjusted by changing the weights, after which they are compensated for by the bias. The sum is calculated by the activation functions and processed to the next layer, as depicted in Figure 6 and expressed as follows:

$$y_j = f(net) = f(\sum_{i=1}^{n} \omega_{ij} x_i + b_j) \tag{13}$$

where $y_j$ is the weighted sum in the $j$th neuron, $xi$ is the input in the $i$th neuron, $w_{ij}$ is the weight between the $i$th and $j$th neurons, $b_j$ is the bias in $j$th neuron and f is the activation function. In this study, sigmoid activation functions were applied to each layer. In Figure 6, the subscript R denotes the number of elements in the input vector, and the subscript S denotes the number of neurons in the layer.

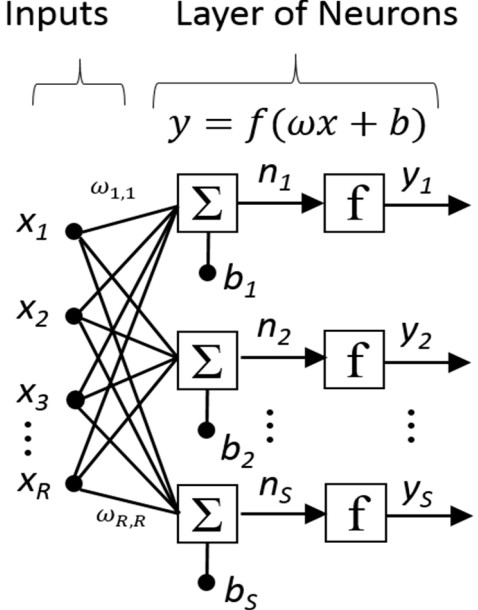

**Figure 6.** Structure of artificial neural network (ANN).

The Levenberg–Marquardt algorithm was chosen and utilized for the back-propagation training, because it is the most efficient algorithm in terms of model prediction efficiency and accuracy [36].

## 4. Experimental Method

### 4.1. Test Specimen

The range of the target concrete strength was set to 21–50 MPa for general construction. All of the slabs were designed with different thicknesses and strengths to obtain a reliable range for test validation. A total of seven specimens were designed with dimensions of 1.5 m × 1.5 m and thicknesses of 0.21–0.30 m, and the slab specimens are shown in Figure 7. They all contained transverse and longitudinal 13 mm-diameter reinforcements with 300 mm spacing at a 50 mm position from the bottom.

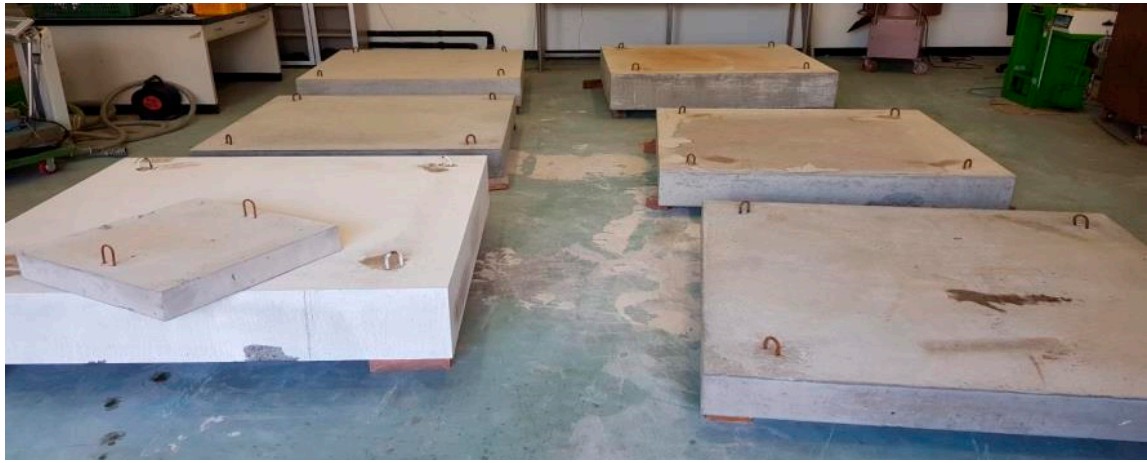

**Figure 7.** Test slabs for the ultrasonic velocity measurements.

According to the test grid as presented in Figure 2, the P-, S-, and R-wave velocities (Vp, Vs, and Vr, respectively) were measured, after which one core was drilled and extracted for each line. Thus, the nine cores on each slab were taken for the compression tests. The dimensions and the target and actual average 28-day strengths of the test slabs are summarized in Table 1 and the concrete mix proportion for each target strength is presented in Table 2.

**Table 1.** Dimensions, target strengths, and actual strengths of specimens.

| Name | Target Strength (MPa) | Width (mm) | Length (mm) | Depth (mm) | Mean Corestrength (MPa) | Age (Test Day) |
|---|---|---|---|---|---|---|
| S24D210 | 27 | 1500 | 1500 | 210 | 25.70 | 28 d |
| S30D400 | 30 | 1500 | 1500 | 400 | 28.31 | 28 d |
| S30D270 | 30 | 1500 | 1500 | 270 | 30.74 | 28 d |
| S30D300 | 30 | 1500 | 1500 | 300 | 32.50 | 28 d |
| S35D210 | 35 | 1500 | 1500 | 210 | 36.63 | 28 d |
| S35D240 | 35 | 1500 | 1500 | 240 | 39.24 | 28 d |
| S50D210 | 50 | 1500 | 1500 | 210 | 51.16 | 28 d |
| S50D240 | 50 | 1500 | 1500 | 240 | 60.71 | 28 d |

**Table 2.** Concrete mix proportions of various target strengths.

| Target Strength (MPa) | Water to Cement Ratio | Cement (kg/m$^3$) | Water (kg/m$^3$) | Fine Aggregate (kg/m$^3$) | Coarse Aggregate (kg/m$^3$) |
|---|---|---|---|---|---|
| 27 | 0.49 | 367 | 180 | 761 | 1055 |
| 30 | 0.45 | 400 | 180 | 734 | 1055 |
| 35 | 0.39 | 462 | 180 | 683 | 1055 |
| 50 | 0.35 | 514 | 180 | 639 | 1055 |

*4.2. Wave Velocity Measurements*

For more accurate prediction and better model development, the compressive strength of one core per column was set as the representative value in that line. The machine learning model was constructed by comparing this value with the average values of Vp, Vs, and Vr for each test line. Tables 3–5 provide the points and line average values of Vp, Vs, and Vr in the S24D210 sample specimen, respectively.

**Table 3.** P-wave velocities (m/s) at test grid points in S24D210 slab (unit: m/s).

| Test Points | Column Line | | | | | | | | |
|---|---|---|---|---|---|---|---|---|---|
| | 1 | 2 | 3 | 4 | 5 | 6 | 7 | 8 | 9 |
| A | 3677 | 3645 | 3639 | 3535 | 3614 | 3633 | 3633 | 3703 | 3770 |
| B | 3559 | 3677 | 3436 | 3565 | 3710 | 3547 | 3783 | 3723 | 3750 |
| C | 3756 | 3639 | 3547 | 3442 | 3697 | 3465 | 3639 | 3633 | 3471 |
| D | 3671 | 3645 | 3414 | 3608 | 3633 | 3529 | 3677 | 3652 | 3671 |
| E | 3756 | 3559 | 3505 | 3881 | 3589 | 3542 | 3589 | 3677 | 3671 |
| F | 3804 | 3482 | 3633 | 3602 | 3710 | 3420 | 3664 | 3633 | 3465 |
| avg. | 3704 | 3608 | 3529 | 3606 | 3659 | 3523 | 3664 | 3670 | 3633 |

**Table 4.** S-wave velocities (m/s) at test grid points in the S24D210 slab (unit: m/s).

| Test Points | Column Line | | | | | | | | |
|---|---|---|---|---|---|---|---|---|---|
| | 1 | 2 | 3 | 4 | 5 | 6 | 7 | 8 | 9 |
| A | 2031 | 2071 | 2050 | 2023 | 1937 | 1982 | 2026 | 2047 | 2023 |
| B | 2023 | 2019 | 2047 | 2047 | 1991 | 2007 | 2048 | 1962 | 2026 |
| C | 2000 | 2014 | 2023 | 1989 | 2030 | 2058 | 2026 | 2047 | 2002 |
| D | 2028 | 2047 | 2047 | 2047 | 2002 | 2035 | 2039 | 2033 | 2050 |
| E | 1958 | 2041 | 1989 | 1966 | 2035 | 2026 | 2071 | 1982 | 261 |
| F | 1994 | 1958 | 2047 | 2047 | 2047 | 2048 | 2026 | 2023 | 2028 |
| avg | 2006 | 2025 | 2034 | 2020 | 2007 | 2026 | 2039 | 2016 | 2032 |

**Table 5.** R-wave velocities (m/s) at test grid points in S24D210 slab (unit: m/s).

| Test Points | Column Line | | | | | | | | |
|---|---|---|---|---|---|---|---|---|---|
| | 1 | 2 | 3 | 4 | 5 | 6 | 7 | 8 | 9 |
| A~F | 1962 | 1948 | 1933 | 1945 | 1952 | 1928 | 1967 | 1959 | 1956 |

Table 6 summarizes the line average velocities of each wave type and the actual core strength values for the S24D210 slab within the strength range of 24~27MPa. The values of Vp, Vs, and Vr were in the typical range of the ultrasonic velocity of normal strength concrete. However, in the case of the S50D210 slab with the a 50 MPa compressive strength, the values of Vp, Vs, and Vr were significantly larger those of the S24D210 as presented in Table 7.

**Table 6.** Line-average P-, S-, and R-wave velocities and test strengths in S24D210 slab.

| Line No. | 1 | 2 | 3 | 4 | 5 | 6 | 7 | 8 | 9 |
|---|---|---|---|---|---|---|---|---|---|
| Vp(m/s) | 3704 | 3608 | 3529 | 3606 | 3659 | 3523 | 3664 | 3670 | 3633 |
| Vs(m/s) | 2006 | 2025 | 2034 | 2020 | 2007 | 2026 | 2039 | 2016 | 2032 |
| Vr(m/s) | 1962 | 1948 | 1933 | 1945 | 1952 | 1928 | 1967 | 1959 | 1956 |
| Test strength (MPa) | 27.40 | 26.00 | 25.20 | 25.60 | 27.10 | 24.30 | 27.20 | 27.30 | 26.00 |

**Table 7.** Line-average P-, S-, and R-wave velocities and test strengths in S50D210 slab.

| Line No. | 1 | 2 | 3 | 4 | 5 | 6 | 7 | 8 | 9 |
|---|---|---|---|---|---|---|---|---|---|
| Vp(m/s) | 4704 | 4541 | 4549 | 4520 | 4549 | 4521 | 4469 | 4519 | 4582 |
| Vs(m/s) | 2565 | 2552 | 2570 | 2575 | 2568 | 2559 | 2547 | 2572 | 2583 |
| Vr(m/s) | 2423 | 2380 | 2389 | 2384 | 2388 | 2378 | 2361 | 2383 | 2402 |
| Core strength (MPa) | 61.50 | 50.80 | 53.90 | 48.40 | 54.60 | 49.80 | 44.10 | 47.60 | 54.70 |

The average values of Vp, Vs, and Vr for all the slabs are listed in Table 8. As the strength increased, the wave velocity increased. The Poisson's ratios were calculated to be in the range of 0.25–0.30 using Equations (1) and (2). These values were slightly larger than the values of general concrete (0.2–0.3), but were reasonable.

**Table 8.** The average values of wave velocity, core strength, and Poisson ratio.

| Div. | Vp (m/s) | Vs (m/s) | Vr (m/s) | Core Strength (MPa) | Poisson Ratio |
|------|----------|----------|----------|---------------------|---------------|
| S24D210 | 3622 | 2023 | 1950 | 25.70 | 0.27 |
| S30D400 | 3563 | 1905 | 1889 | 28.31 | 0.30 |
| S30D270 | 3848 | 2159 | 2058 | 30.74 | 0.27 |
| S30D300 | 3795 | 2178 | 2054 | 32.50 | 0.25 |
| S35D210 | 4044 | 2179 | 2112 | 39.24 | 0.30 |
| S35D240 | 3925 | 2214 | 2099 | 36.63 | 0.27 |
| S50D210 | 4642 | 2543 | 2400 | 60.71 | 0.29 |
| S50D240 | 4550 | 2566 | 2388 | 51.16 | 0.27 |

Figure 8 shows the relationship between the P-, S-, and R-wave velocities and the core strengths for all the specimens. All the wave velocities exhibited linear correlations with the strength, and the correlation coefficients for the P-, R-, and S-wave velocities were 0.947, 0.905, and 0.831, respectively in order. The mean squared errors (MSEs) between the actual strengths and those predicted by the regression for all of the slabs are presented in Table 9. In most cases, the differences between the measured and predicted strengths were relatively small. However, in the case of sample S27D400, the differences between the values of Vp, Vs, and Vr were large, and an MSE of 47.02 was obtained for Vs. When the amount of aggregate was less than that of normal concrete or the distribution was uneven, the elasticity was small, which affected the ultrasonic velocity.

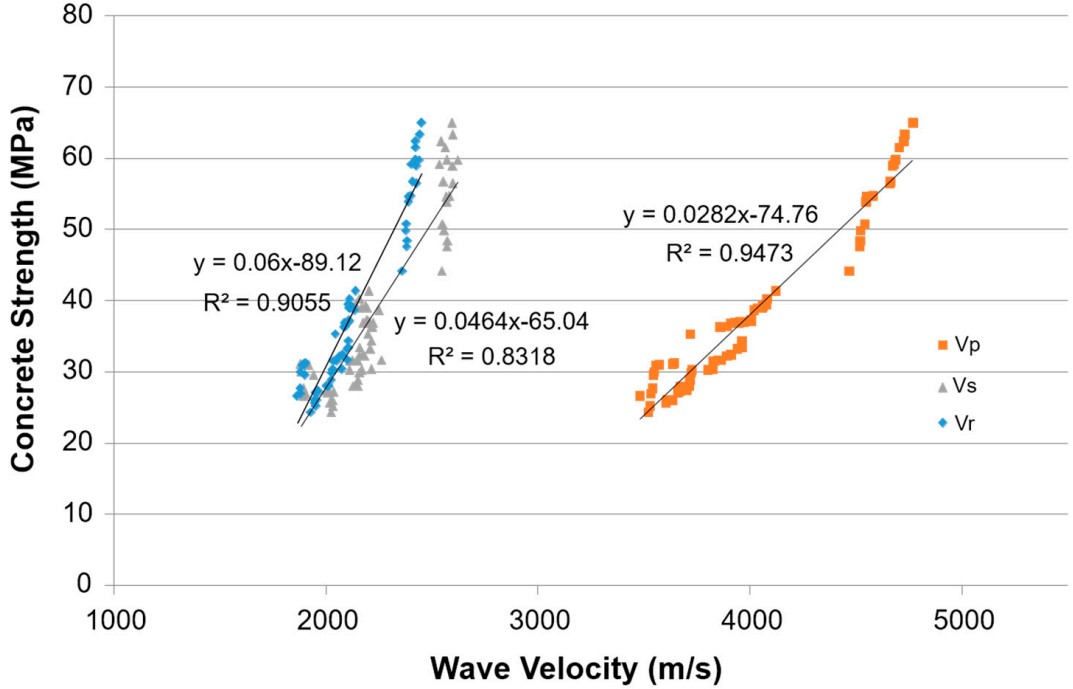

**Figure 8.** Relationship of P-, S-, and R-wave velocities with concrete strength.

**Table 9.** Mean squared error of regressions for P-, S-, and R-wave velocities.

| Div. | Regression Method | | | Core Strength (MPa) | Mean Square Error | | |
|---|---|---|---|---|---|---|---|
| | Estimation by Vp (MPa) | Estimation by Vs (MPa) | Estimation by Vr (MPa) | | Estimation by Vp | Estimation by Vs | Estimation by Vr |
| S24D210 | 26.65 | 28.00 | 27.89 | 26.23 | 0.17 | 3.13 | 2.74 |
| S27D400 | 25.00 | 22.59 | 24.20 | 29.44 | 19.73 | 47.02 | 27.52 |
| S30D270 | 32.99 | 34.27 | 34.39 | 31.66 | 1.79 | 6.85 | 7.46 |
| S30D300 | 31.50 | 35.17 | 34.11 | 30.43 | 1.13 | 22.40 | 13.53 |
| S35D210 | 38.47 | 35.20 | 37.62 | 38.81 | 0.12 | 13.02 | 1.42 |
| S35D240 | 35.14 | 36.79 | 36.79 | 37.07 | 3.72 | 0.08 | 0.08 |
| S50D210 | 55.20 | 51.92 | 54.85 | 57.82 | 6.85 | 34.84 | 8.83 |
| S50D240 | 52.65 | 52.98 | 54.13 | 51.71 | 0.88 | 1.62 | 5.87 |
| | | | | avg | 4.30 | 16.12 | 8.43 |

## 5. Analysis of Results

The optimum parameters were obtained for the SVM and ANN regression algorithms using an optimum search approach corresponding to the best generalized performance for the trained model based on measured performance criteria, such as the mean square error, as listed in Table 10.

**Table 10.** Optimum parameters of support vector machine (SVM) and ANN models with P-wave velocity.

| SVM | | ANN | |
|---|---|---|---|
| Parameter | Value | Parameter | Value |
| Bias | 42.75 | Hidden Layers | 1 |
| Box constraint | 9.4885 | Hidden Neurons | 50 |
| Epsilon | 0.9489 | Training function | trainlm |
| Number of iterations | 64 | Epoch | 5 |
| Kernel | Gaussian | | |

For the SVM, the model was built using all the data (three types of ultrasonic velocities), and the errors with the target values (core strength) were analyzed for accuracy evaluation. For the ANN, 70% of the total data was used to construct the model, 15% was used for data verification, and 15% was used for testing. The ANN model constructed in this manner was compared with the SVM for the entire data set for comparative evaluation.

The prediction accuracies of the SVM and ANN models were evaluated using the correlation coefficient (R), mean absolute error (MAE), mean relative error, and mean squared error (MSE), defined as follows:

$$R = \frac{n \sum_{i=1}^{n} y_i \cdot \hat{y}_i - \left(\sum_{i=1}^{n} y_i\right)\left(\sum_{i=1}^{n} \hat{y}_i\right)}{\sqrt{\sum_{i=1}^{n} y_i^2 - \left(\sum y_i\right)^2} \sqrt{\sum_{i=1}^{n} \hat{y}_i^2 - \left(\sum \hat{y}_i\right)^2}} \tag{14}$$

$$MAE = \frac{1}{n} \sum_{i=1}^{n} |y_i - \hat{y}_i| \tag{15}$$

$$MRE = \frac{1}{n} \sum_{i=1}^{n} \frac{|y_i - \hat{y}_i|}{y_i} \tag{16}$$

$$MSE = \frac{\sum_{i=1}^{n} (y_i - \hat{y}_i)^2}{n} \tag{17}$$

where $y_i$ is the target value, $\hat{y}_i$ is the predicted value, and the n is the number of data.

For the comparison, the SVM and ANN models were constructed with different input variables: (a) P-wave velocity, (b) P- and S-wave velocities, and (c) P-, S-, and R-wave velocities.

The output variable was the actual core strength for all the models. After training with the 72 test sample data, all the samples in the testing data set were used to examine the model prediction accuracy. The model prediction results are summarized in Table 11.

**Table 11.** SVM and ANN model prediction accuracy.

| Method | SVM | | | ANN | | |
|---|---|---|---|---|---|---|
| Input Variable | Vp | Vp, Vs | Vp, Vs, Vr | Vp | Vp, Vs | Vp, Vs, Vr |
| Correlation coefficient(R) | 0.985 | 0.991 | 0.994 | 0.984 | 0.989 | 0.993 |
| MAE | 1.581 | 1.218 | 1.072 | 1.666 | 1.359 | 0.987 |
| MRE | 0.044 | 0.033 | 0.029 | 0.048 | 0.037 | 0.027 |
| MSE | 3.864 | 2.404 | 1.719 | 4.071 | 2.832 | 1.790 |

As summarized in Table 11, the correlation did not increase significantly as the number of input variables increased from one to three. However, as the number of variables increased, the prediction accuracy improved more than the correlation. Overall, the SVM and ANN models with three input variables (P-, S-, and R-wave velocities) yielded the best prediction results in terms of the MAE, MRE, and MSE. For the MAE, which mainly measures the performance of the machine learning, the error was reduced by more than 50% when three variables were used instead of one variable in the SVM and ANN models. Compared with the linear regression, there was no significant difference between the SVM and ANN with one variable (P-wave velocity). Thus, more types of ultrasonic velocities, rather than the predictive method, can provide a more accurate prediction.

Relative comparisons between the SVM and ANN showed a 4.13% improvement in the MSE when three variables were used. Figure 9 shows the correlation between the actual and predicted strengths. Both SVM and ANN can make good predictions, also the SVM was more highly correlated with the actual strength than the ANN, and it was more concentrated on the equality line. Figure 10 shows the ratio of the predicted value to the actual value. The data from the SVM model was closer to a ratio of 1 than that of the ANN model, and the data was highly concentrated.

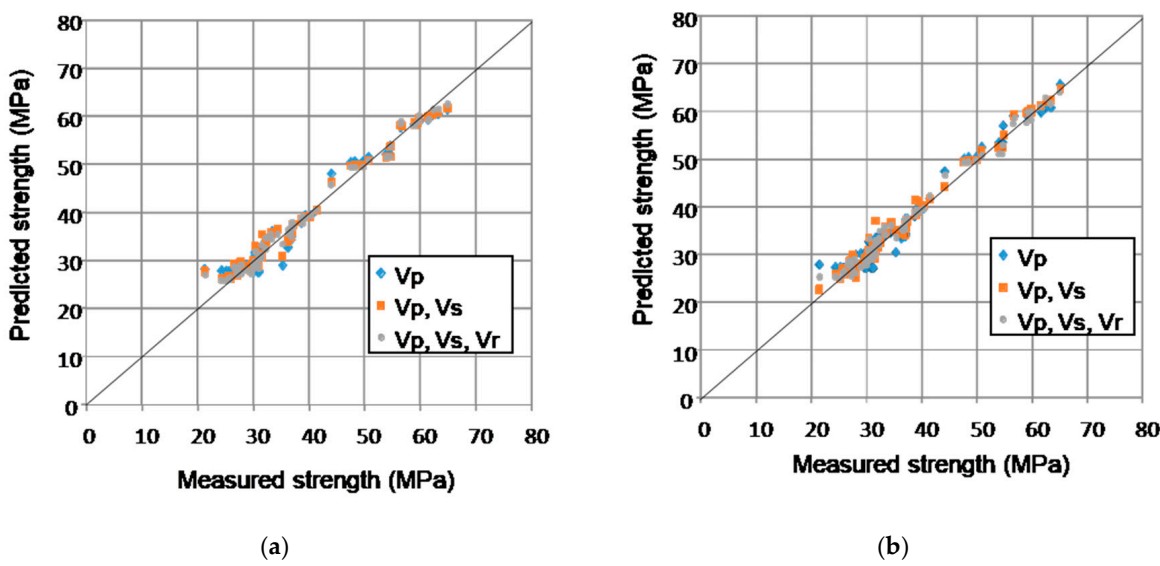

(**a**)                (**b**)

**Figure 9.** Correlation of predicted to measured concrete strength using the SVM and ANN models. (**a**) SVM; (**b**) ANN.

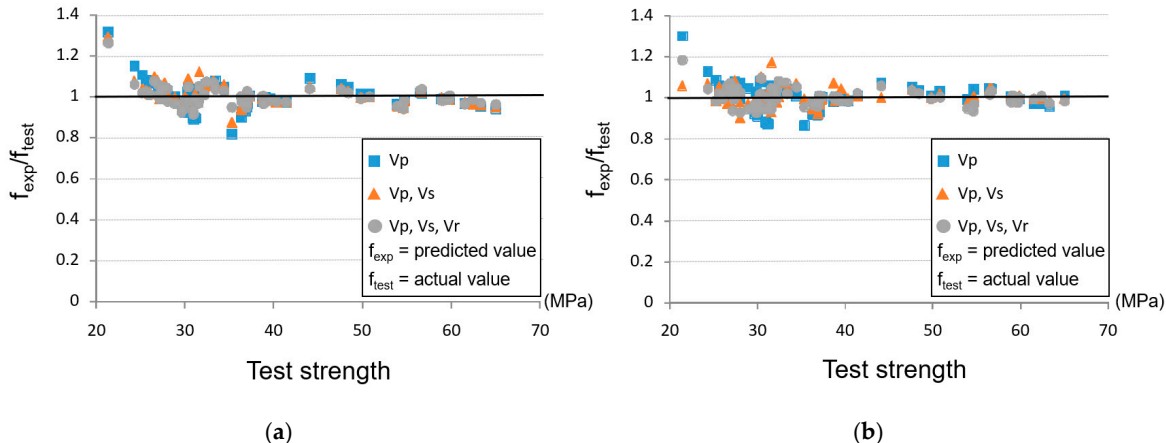

(**a**)　　　　　　　　　　　　　　(**b**)

**Figure 10.** Distribution of predicted strength to test strength ratios. (**a**) SVM; (**b**) ANN.

The accessibility of predicted strength by SVM and ANN into the target strength is presented in Figure 11, and there was less data variance for the SVM model than the ANN model. Understanding the nonlinearity of the predicted model was difficult owing to the lack of data in the 50–60 MPa range. The predicted and measured strengths exhibited abrupt changes above 50 MPa, and thus, the ability of the model to simulate the nonlinearity at normal and high strengths is important.

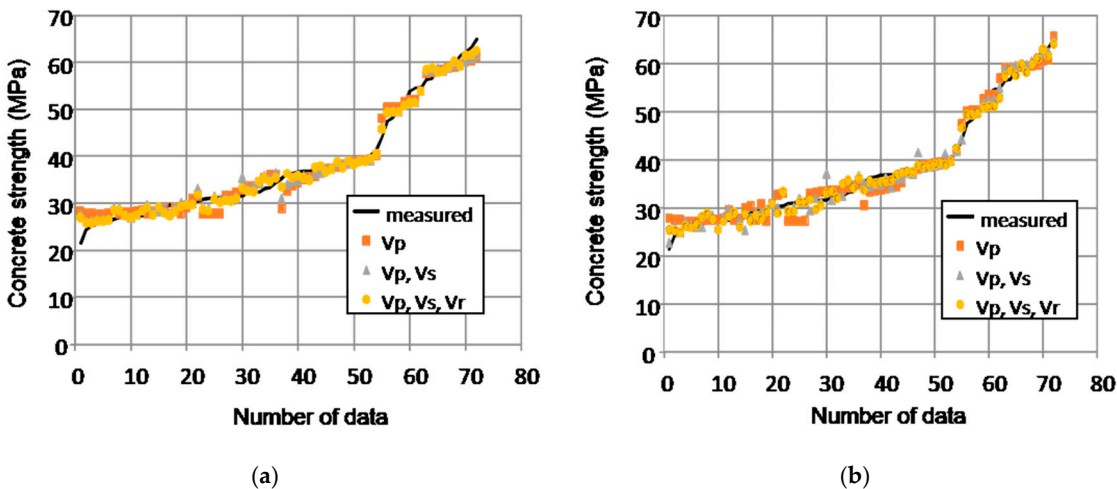

(**a**)　　　　　　　　　　　　　　(**b**)

**Figure 11.** Tracking of measured concrete strength by SVM and ANN. (**a**) SVM; (**b**) ANN.

In Figure 12, the empirical cumulative distribution function provided by MATLAB was applied to the predicted data, and a ±25% deviation was plotted. First, it was confirmed that below 50 MPa, the data were below the distribution curve, and above 50 MPa, they were above this curve. This was because the velocities of the ultrasonic waves changed significantly from 50 MPa because of the change in the mechanical properties between the normal and high strengths.

The convergence efficiency and prediction accuracy of the ANN method for the data variables are shown in Figure 13. It was confirmed that convergence was faster, and the MSE (performance) decreased as the number of ultrasonic velocity types increased. Although the prediction accuracy for the validation and test data sets is large due to the limited number of data and nonlinearity of concrete, the differences of prediction in validation and test data sets are greatly reduced and the accuracy is greatly increased with Vp, Vs, and Vr.

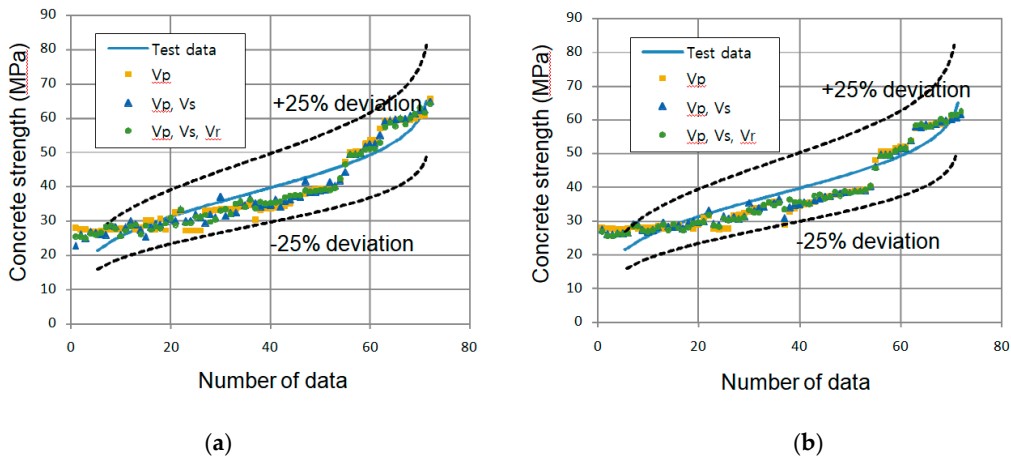

**Figure 12.** Predicted data distributions by SVM and ANN within a ±25% deviation from the empirical cumulative distribution curve of the test data. (**a**) SVM; (**b**) ANN.

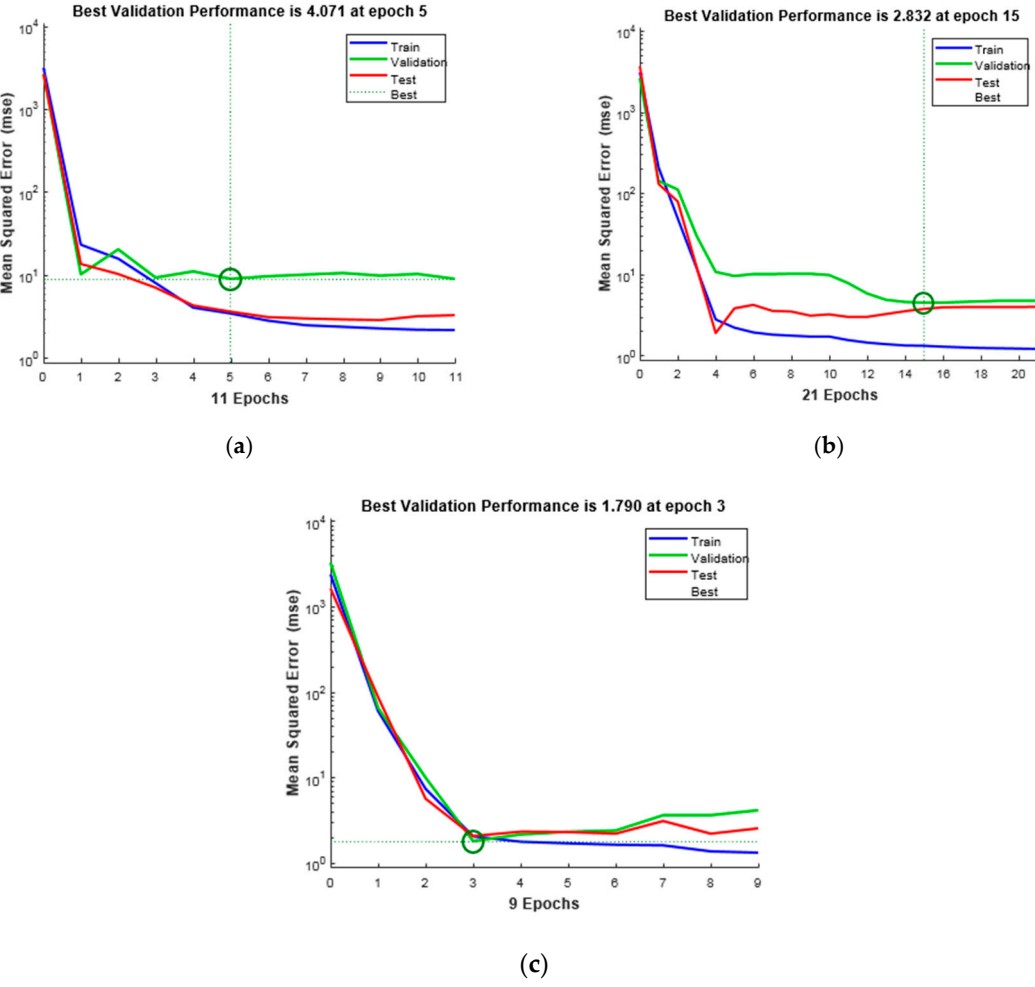

**Figure 13.** ANN parameter optimization using algorithm convergence. (**a**) P-wave; (**b**) P-, S- waves; (**c**) P-, S-, R- waves.

This study analyzed the performances and accuracies of the SVM and ANN models with three types of ultrasonic velocities. As in previous studies, the SVM was more accurate than the ANN because of the inherent drawbacks of the ANN, such as the slow convergence, less generalizable performance, tendency to find only local minima, and over-fitting problems. The generalization of the developed model was difficult owing to the nonlinearity in the normal- and high-strength concrete,

and it may be more reasonable to use individual models for these concrete types. Moreover, as the concrete strength decreased (in the low strength concrete), various factors, such as the type of aggregate and cement, water to cement ratio, and aggregate interface specificity, must be involved in addition to the ultrasonic velocity. Nevertheless, the mix design information of an existing structure is generally not easily obtained. In this sense, the measurement of Vp, Vs, and Vr can predict the actual strength more accurately even though the economic cost including the device and the human effort for the proposed measurements is more expensive than the traditional methods. Furthermore, these velocities can capture the nonlinearities present for normal- and high-strength concretes.

## 6. Conclusions

A predictive model of concrete strength using measurements of the P-, S-, and R-wave velocities was proposed as a non-destructive method using support vector machine (SVM) and artificial neural network (ANN) models. The strength prediction models were developed using the correlation between the ultrasonic velocities and core strengths using six slabs in a strength range of 24–60 MPa. The SVM and ANN models predicted the strengths with much higher accuracies than the traditional linear regression, and the following conclusions were drawn.

- In the prediction of concrete strength, the predictive models that used three types of ultrasonic velocities were more accurate than the models that used only one or two velocities, and they converged more quickly with smaller errors.
- The SVM model was able to obtain more accurate predictions than the ANN because of the less generalizable performances and over-fitting issues of ANNs.
- A slight change in the ultrasonic velocity was observed at around 50 MPa, and individual models are required for normal- and high-strength concretes to obtain higher accuracy.
- For strengths less than 30 MPa, the differences and dispersion of the ultrasonic velocities and core strengths were large, and various factors, such as the type, size, and distribution of the aggregate and cement as well as the water to cement ratio, should be considered to obtain more accurate predictions.

**Author Contributions:** J.Y.P. conceived and designed the experiments; Y.G.Y. and T.K.O. performed the experiments and analyzed the data; J.Y.P. contributed device/analysis tools; Y.G.Y. and T.K.O. wrote the paper.

**Acknowledgments:** This research was supported by Basic Science Research Program through the National Research Foundation of Korea (NRF) funded by the Ministry of Education (No.2018R1D1A1A02085377). This work was supported by Research Assistance Program (2019) in the Incheon National University.

**Conflicts of Interest:** The authors declare no conflict of interest.

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
