# Peer review of "Prediction of Concrete Strength with P-, S-, R-Wave Velocities by Support Vector Machine (SVM) and Artificial Neural Network (ANN)"

_applsci, doi:10.3390/app9194053_

Round 1

Reviewer 1 Report

Review: applsci-583075-peer-review-v1

Prediction of concrete strength with P-, S-, R-wave 2 velocities by Support Vector Machine (SVM) and 3 Artificial Neural Network (ANN)

The authors estimated concrete compressive strength using two predictive models: a) Support Vector Machine (SVM) and b) artificial neural network (ANN). Database to train the methods obtained from a series of laboratory testing (6 slabs). The dataset to train the SVM and ANN includes the velocities of P-, S- and Rayleigh waves measured using Ultrasonic Nondestructive Testing.

Comments:

The paper needs proofreading and improvement in language. Line 135: (1-10) or (1-9)? Line 298: Table 2 should be inserted again. The numbers are not clear. Line 311: Table 7 à Is S30D400 correct type? Shouldn’t be S27D400? Line 320: Figure 8 à Firstly, the y-axis margin is changing in the context of paper which makes comparison of different methods and regression difficult. It is better to stick to one range for example 0-80 MPa. Secondly, As it is mentioned in the conclusion, a change in the ultrasonic wave velocity is observed around 50MPa. How about using two linear regression above and below that point? Or using a second order regression? Line 357: What is this %4.30 improvement? Compared to what? Line 365 Figure 9 à Lack of this graph for linear regression method for better comparison. Line 368: Figure 10 à better y axis label. Considering these graphs, using only Vp on your results gives satisfying accurate results compared to using all three Vp, Vs, Vr which adds to the complexity of measuring S- and R- waves velocities compared to a simple pulse echo test. By assuming some points as outliers, how do you think about the results using only Vp. In addition to that, you used indirect method for your pulse echo which we know that you get attenuated signal. What if you use direct method? Wouldn't get satisfying results? Line 369: What is data accessibility into the actual strength? Line 393: Figure 13 à These graphs have major problems. Do graphs (a) and (b) show 4.071 and 2.832 MSE for validation dataset respectively? They do not show the mentioned value in the graph and the error is much larger. why does graph (a) show very different MSE between validation and test dataset? Did you try more hidden layers for your NN? Did you consider the strength of your material in making the training dataset or just randomly selecting %70 of the dataset? When the waves velocities show two different behavior for concrete with higher and lower that 50 MPa, considering two models for above and below that margin might enhance the results significantly.

To conclude, I find that, apart from the issues I’ve elaborated above, the paper might be fit for publication after addressing the issues.

Reviewer 2 Report

The authors propose in this work the prediction model of concrete strength by measurement of P-, S- and R-wave velocities as a non-destructive method associated with SVM and ANN. The SVM and ANN models predicted the strength with much higher accuracy than the traditional linear regression. This work can be potentially be published in MDPI pending minor revisions as listed below:

Lines 11-12 should be removed.

Line 13: ‘A mechanical waves’ should change to ‘A mechanical wave’.

Fig. 1(b) is not clear of what is shown. I would like to see a block diagram instead showing the procedure of the P-wave measurement.

Similarly for Fig. 3(b).

Can authors use more than one hidden layer to further optimize the system?

What will happened to the results if a different nonlinear activation function is used (e.g. sigmoid function)?

Please fix in the x-axis of Fig. 10 the ‘MPa’.

There is nothing written in author contributions.

Will the implementation of another machine learning algorithm such as clustering (i.e. K-means, fuzzy-logic, etc.) give any benefits over SVM or ANN?

Does the complexity of the chosen machine learning algorithm matters in this application?

What is the training, testing and validation time?

Reviewer 3 Report

The study proposed the prediction model of concrete strength by measurement different kinds of wave velocities as a non-destructive method associated with a support vector machine (SVM) and Artificial Neural Network (ANN). Estimating concrete compressive strength by non-destructive methods is an important issue in concrete technology. The authors used advanced methods of prediction compressive strength more accurate.

Additionally, the attempt to increase the accuracy of the UPV method by taking into account S and R-waves is an original approach to this commonly used method. Also conducted tests of the concrete slabs are an original experiment. The description of the tests carried out shows that the authors planned the tests and analysis of their results very carefully.

Authors demonstrated that when SVM or ANN are used, as the number of variables increases, the prediction accuracy improves more than correlation. Also, the predicted values are closer to measured, comparing to the linear regression method.

The authors noted that for concretes with strength above 50 MPa, the method overstates the results. Perhaps this would be justified by the high strength concrete composition? Unfortunately, the authors did not provide the composition of the mixtures tested.

There is also a lack of information about the reinforcement of tested slabs.

The figures are readable, although the resolution of some illustrations in the manuscript I received for review could be a bit higher.

Some of the minor mistakes:

Line 101 - what do authors mean by "R-wave is very sensitive on the concrete." line 101 and 102 contain the same sentence "S-wave is not affected by water" line 103 - is information in that paragraph result from the author's knowledge? are they supported by research results? if so, please cite relevant publications. 2 - there is a lack of units Line 193-195 - symbols in the formulas should be described. 8 - MPa instead of Mpa Line 343 - lack of “.” At the end of the paragraph.
